# Shaping embodied agent behavior with activity-context priors from egocentric video

**Tushar Nagarajan**
UT Austin and Facebook AI Research
`tushar.nagarajan@utexas.edu`

**Kristen Grauman**
UT Austin and Facebook AI Research
`grauman@cs.utexas.edu`

## Abstract

Complex physical tasks entail a sequence of object interactions, each with its own preconditions—which can be difficult for robotic agents to learn efficiently solely through their own experience. We introduce an approach to discover *activity-context* priors from in-the-wild egocentric video captured with human worn cameras. For a given object, an activity-context prior represents the set of other compatible objects that are required for activities to succeed (e.g., a knife and cutting board brought together with a tomato are conducive to *cutting*). We encode our video-based prior as an auxiliary reward function that encourages an agent to bring compatible objects together before attempting an interaction. In this way, our model translates everyday human experience into embodied agent skills. We demonstrate our idea using egocentric EPIC-Kitchens video of people performing unscripted kitchen activities to benefit virtual household robotic agents performing various complex tasks in AI2-iTHOR, significantly accelerating agent learning. Project page: http://vision.cs.utexas.edu/projects/ego-rewards/

## 1   Introduction

Embodied AI agents that are capable of moving around and interacting with objects in human spaces promise important practical applications for home service robots, ranging from agents that can search for misplaced items, to agents that can cook entire meals. The pursuit of such agents has driven exciting new research in visual semantic planning [69], instruction following [59], and object rearrangement [3, 30], typically supported by advanced simulators [46, 33, 58, 22] where policies may be learned quickly and safely before potentially transferring to real robots. In such tasks, an agent aims to perform a sequence of actions that will transform the visual environment from an initial state to a goal state. This in turn requires jointly learning behaviors for both *navigation*, to move from one place to another, and object *interaction*, to manipulate objects and modify the environment (e.g., pick-up objects, use tools and objects together, turn-on lights).

A key challenge is that changing the state of the environment involves context-sensitive actions that depend on both the agent and environment's current state—what the agent is holding, what other objects are present nearby, and what their properties are. For example, to wash a plate, a plate must be in the sink *before* the agent toggles-on the faucet; to slice an apple the agent must first be holding a knife, and the apple must not already be sliced. Understanding these conditions is critical for efficient learning and planning: the agent must first bring the environment into the proper precondition state before attempting to perform a given activity with objects. We refer to these states as *activity-contexts*.

Despite its importance, current approaches do not explicitly model the activity-context. Pure reinforcement learning (RL) approaches directly search for goal states without considering preconditions. This requires a large number of trials for the agent to chance upon suitable configurations, and leads to poor sample efficiency or learning failures [69, 59]. Instead, most methods resort to collecting expert demonstrations to train imitation learning (IL) agents and optionally finetune with

35th Conference on Neural Information Processing Systems (NeurIPS 2021).

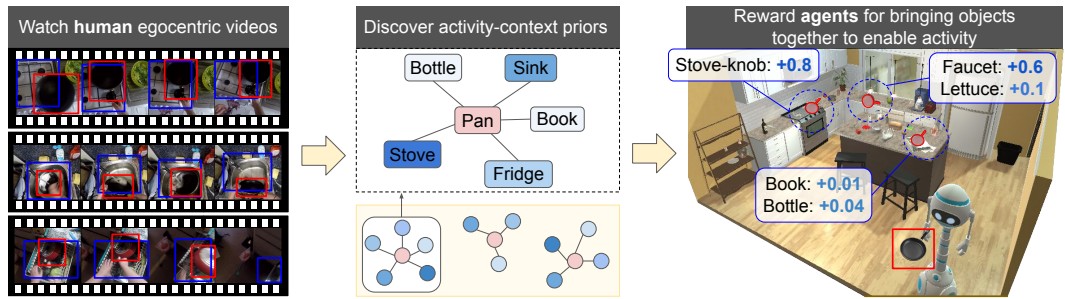

Figure 1: **Main idea. Left and middle panel:** We discover *activity-contexts* for objects directly from egocentric video of human activity. A given object's activity-context goes beyond "what objects are found together" to capture the likelihood that each other object in the environment participates in activities involving it (i.e., "what objects together enable action"). **Right panel:** Our approach guides agents to bring compatible objects—objects with high likelihood—together to enable activities. For example, bringing a pan to the sink increases the value of faucet interactions, but bringing it to the table has little effect on interactions with a book.

RL [69, 59, 31, 15]. While demonstrations may implicitly reveal activity-contexts, they are cumbersome to collect; they require expert teleoperation using specialized hardware (VR, MoCap), often in artificial lab settings [68, 49, 28], and need to be collected independently for each new task.

Rather than solicit demonstrations, we propose to learn activity-contexts from real-world, egocentric video of people performing daily life activities. Humans understand activities from years of experience, and can effortlessly bring even a novel environment to new, appropriate configurations for interaction-heavy tasks. Egocentric or "first-person" video recorded with a wearable camera puts actions and objects manipulated by a person at the forefront, offering an immediate window of this expertise in action in its natural habitat.

We present a reinforcement learning approach that infers activity-context conditions directly from people's first-hand experience and transfers them to embodied agents to improve interaction policy learning. Specifically, we 1) train visual models to detect how humans *prepare* their environment for activities from egocentric video, and 2) develop a novel auxiliary reward function that encourages agents to seek out similar activity-context states. For example, by observing that people frequently carry pans to sinks (to clean them) or stoves (to cook their contents), our model rewards agents for prioritizing interactions with faucets or stove-knobs when pots or pans are nearby. As a result, this incentivizes agents to transport relevant objects to compatible locations *before* attempting interactions, which accelerates learning. See Fig. 1.

Importantly, our goal is not direct imitation. Our insight is that while humans and embodied agents have very different action spaces and bodies, they operate in similar environments where the underlying conditions about what state the environment must be in before trying to modify it are strongly aligned. Our goal is thus to guide an agent's exploratory interactions towards these potential progress-enabling states as it learns a new task. Moreover, because our training videos are passively collected by human camera-wearers and capture a wide array of daily actions, they help build a general visual prior for human activity-contexts, while side-stepping the heavy requirements of collecting IL demonstrations for each individual task of interest.

Our experiments demonstrate the value of learning from egocentric videos from *humans* (in EPIC-Kitchens [13]) to train visual semantic planning *agents* (in AI2-iTHOR [33]). Our video model relates objects based on goal-oriented actions (e.g., knives used to cut potatoes) rather than spatial co-occurrences (e.g., knives are found near spoons) [66, 11, 48, 65] or semantic similarity (e.g., potatoes are like tomatoes) [66]. Our agents outperform strong exploration methods and state-of-the-art embodied policies on multi-step interaction tasks (e.g., storing cutlery, cleaning dishes), improving absolute success rates by up to 12% on the most complex tasks. Our approach learns policies faster, generalizes to unseen environments, and greatly improves success rates on difficult instances.

## 2 Related Work

**Interaction in 3D environments** Recent embodied AI work leverages simulated environments [46, 52, 58, 33, 22, 58] to build agents for interaction-heavy tasks like visual semantic planning [69], interactive question answering [25], instruction following [59], and object rearrangement [3, 31].

Prior approaches use imitation learning (IL) based pre-training to improve sample efficiency, but at the expense of collecting expert in-domain demonstrations individually for each task [69, 15, 31, 59]. Instead, we leverage readily available human egocentric video in place of costly IL demonstrations, and we propose an RL approach that benefits from the discovered human activity-priors.

**Exploration for navigation and interaction** Exploration strategies for visual navigation encourage efficient policy learning by rewarding agents for covering area [12, 10, 17], visiting new states [51, 4], expanding the frontier [47], anticipating maps [50], or via intrinsic motivation [42, 6]. For grasping, prior work studies curiosity [27] and disagreement [43] as intrinsic motivation. Beyond navigation and grasping, interaction exploration [40] rewards agents for successfully attempting new object interactions (take potato, open fridge, etc.). In contrast, our agents are rewarded for achieving compatible activity-contexts learned from human egocentric video. Our model incentivizes action sequences that are aligned with activities (e.g., putting a plate in the sink *before* turning-on the faucet), as opposed to arbitrary actions (e.g., turning the faucet on and then immediately off).

**Learning from passive video for embodied agents** Thus far, video plays a limited role in learning for embodied AI, primarily as a source of demonstrations for imitation learning. Prior work learns dynamics models for behavior cloning [57, 67, 44, 54, 53] or crafts reward functions that encourage visiting expert states [1, 16, 55, 36]. Beyond imitation, recent navigation work "re-labels" video frames with pseudo-action labels to learn navigation subroutines [9] or action-conditioned value functions [34]; however, they use videos generated from simulation or from intentionally recorded real-estate tours. In contrast, our work is the first to use free-form human-generated video captured in the real world to learn priors for object interactions. Our priors are not tied to specific goals (as in behavior cloning) and are cast as general purpose auxiliary rewards to encourage efficient RL.

**Learning about human actions from video** Substantial work in computer vision explores models for human action recognition in video [70, 62, 8, 19], including analyzing hand-object interactions [56, 2, 61, 7, 24] and egocentric video understanding [35, 21, 38, 32, 13, 26]. More closely related to our work, visual affordance models derived from video can detect likely places for actions to occur [39, 18, 41], such as where to grasp a frying pan, and object saliency models can identify human-useable objects in egocentric video [14, 5, 20]. These methods learn important concepts from human video, but do not consider their use for embodied agent action, as we propose.

**Semantic priors for embodied agents** Human-centric environments contain useful semantic and structural regularities. Prior work exploits spatial relationships between objects [66, 11, 48] and room layouts [65] to improve navigation (e.g., beds tend to be in bedrooms; kitchens tend to be near dining rooms). Recent work learns visual priors from agent experience (not human video) to understand environment traversability [47], affordances [40], or object properties like mass [37]. These priors encode static properties of objects or geometric relationships between them by learning from co-occurrences in static images. In contrast, our formulation encodes information about objects *in action* from video of humans using objects to answer *what objects should be brought together to enable interactions*, rather than *what objects are typically co-located*. In addition, unlike previous models that require substantial online agent experience to learn priors, we use readily available egocentric video of human activity.

## 3 Approach

Our goal is to train agents to efficiently solve interaction-heavy tasks. Training reinforcement learning (RL) agents is difficult due to sparse task rewards, large search spaces, and context-sensitive actions that depend on what the agent is holding and where other objects are. We propose an auxiliary reward derived from egocentric videos of daily human activity that encourages agents to configure the environment in ways that facilitate successful activities. For example, watching humans wash spoons in the sink suggests that the sink is a good location to bring utensils to before interacting with the nearby faucet or dish soap.

In the following, we begin by defining the visual semantic planning task (Sec. 3.1). Then, we show how to infer activity-context priors from egocentric video (Sec. 3.2) and how to translate those priors to an embodied agent in simulation (Sec. 3.3). We then describe our approach to encode the priors as a dense reward (Sec. 3.4). Finally, we present our policy learning architecture (Sec. 3.5).

## 3.1 Visual semantic planning

We aim to train an agent to bring the environment into a particular goal state specified by the task $\tau$. Agents can perform navigation actions $\mathcal{A}_N$ (e.g., move forward, rotate left/right) and object interactions $\mathcal{A}_I$ (e.g., take/put, open, toggle) involving an object from set $\mathcal{O}$.

Each task is set up as a partially observable Markov decision process. The agent is spawned at an initial state $s_0$. At each time step $t$, the agent in state $s_t$ receives an observation $(x_t, \theta_t, h_t)$ containing the RGB egocentric view, the agent's current pose, and the currently held object (if any). The agent executes an action on an object $o_t$, $(a_t, o_t) \sim \{\mathcal{A}_N \bigcup \mathcal{A}_I\}$, and receives a reward $r_t \sim \mathcal{R}_\tau(s_t, a_t, o_t, s_{t+1})$. For navigation actions, the interaction target $o_t$ is *null*. A recurrent network encodes the agent's observation history over time to arrive at the state representation (detailed below).

A task reward is provided if the agent brings its environment to a goal state $g_\tau$. For example, in the "Clean object" task in our experiments, a *washable* object must be inside the sink, and the faucet must be *toggled-on* for success. A positive reward is given for goal completion, while a small negative reward is given at each time-step to incentivize quick task completion:

$$R_\tau(s_t, a_t, o_t, s_{t+1}) = \begin{cases} 10 & \text{if } g_\tau \text{ satisfied,} \\ -0.01 & \text{otherwise.} \end{cases} \tag{1}$$

The goal is to learn a policy $\pi_\tau$ that maximizes this reward over an episode of length $T$.

## 3.2 Human activity-context from egocentric video

Learning interaction policies requires an understanding of how the environment needs to be configured—where the agent needs to be, and what objects need to be present there—before goals are completed. We infer this directly from a passively collected dataset of human egocentric videos.

The dataset consists of a set of clips $v \in \mathcal{V}$ where each clip captures a person performing an interaction with some object from a video object vocabulary $\mathcal{O}_V$ (e.g., "slice tomato", "pour kettle"). Our model uses the clip's frames only, not the interaction label. Our approach first extracts activity-contexts from each training video frame, and then aggregates their statistics over all training clips to produce an *inter-object activity-context compatibility function*, as follows.

For each frame $f_t$ in an egocentric video clip $v$, we use an off-the-shelf hand-object interaction model [56] that detects active objects—manipulated objects in contact with hands—resulting in a set of class-agnostic bounding boxes. We associate object labels to these boxes using a pre-trained object instance detection model. Specifically, we transfer labels from high-confidence instance detections to active object boxes with large overlap, namely, where intersection over union (IoU) $> 0.5$, resulting in a set of active object boxes and corresponding class labels $\mathcal{D}(f_t) = \{(b_0, o_0)..(b_N, o_N)\}$, where $b_i$ denotes box coordinates and $o_i \in \mathcal{O}_V$. These instances represent objects that are directly involved in the activity, ignoring background objects that are incidentally visible but not interacted with by hands. Detection model details are in Supp.

We infer frame $f_t$'s activity-context $AC(f_t)$ from the Cartesian product of active objects.

$$AC(f_t) = \{(o_i, o_j) \mid o_i, o_j \in \mathcal{D}(f_t) \times \mathcal{D}(f_t)\}, \tag{2}$$

where $o_i \neq o_j$, and each $o_i$ is an object that can be held and moved to different locations, as opposed to a fixed object like a refrigerator or sink. We include a *null* object token to consider cases when the agent visits locations empty-handed. Fig. 2 (left) shows examples.

Each $(o_i, o_j)$ pair represents a particular object $o_i$ and a corresponding activity-context object (ACO) $o_j$—an object that is used with it in an activity. These include movable objects (e.g., tools like knives and cutting-boards in "slice tomato"), receptacles (e.g., kettles and cups in "pour water"), and fixed environment locations (e.g., sinks, faucets in "wash spoon"). This is in contrast to an object's *affordance*, which defines object properties in isolation (e.g. tomatoes are *sliceable* whether a knife is held or not, spoons are *washable* even when they are inside drawers).

Finally, we define the inter-object activity-context compatibility score $\phi(o_i, o_j)$ as follows:

$$\phi(o_i, o_j) = \frac{\sum\limits_{v \in \mathcal{V}} S_v(o_i, o_j)}{\sum\limits_{v \in \mathcal{V}} \sum\limits_{o_k \neq o_i} S_v(o_i, o_k)}, \tag{3}$$

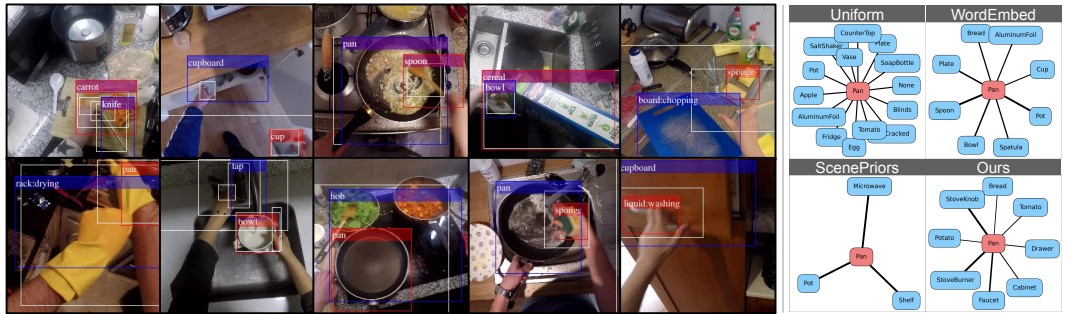

Figure 2: **Left: Detected ACOs from EPIC videos.** White boxes: active object detections. Red/blue boxes: sampled (object, ACO) tuple (Equation 2). Last column shows failure cases—sponge not held (top), liquid:washing false positive (bottom). **Right: ACO compatibility scores in THOR.** Red: active object, blue: top corresponding ACOs. Edge thickness indicates compatibility (Equation 3). Our approach (bottom right) prioritizes object relationships for "Pan" that are relevant for activity (e.g., StoveBurner, StoveKnob) over semantically similar or spatially co-occurring objects (e.g., Pot, Shelf in top-right, bottom-left respectively).

where $S_v(o_i, o_j)$ measures the fraction of frames of video $v$ where $(o_i, o_j) \in AC(f_t)$. By aggregating across clips, we capture dominant object relationships and reduce the effect of object detection errors.

Recall that these relationships are derived from video frames of objects *in action* rather than arbitrary video frames or static images, and are thus strongly tied to human activity, not static scene properties. In other words, $\phi(o_i, o_j)$ measures how likely $o_i$ is *brought together with* $o_j$ during an activity (e.g., knives and cutting boards are used to cut fruits) rather than how likely they generally co-occur in space (e.g., spoons and knives are stored together, but neither enables the action of the other).

### 3.3 Translating to activity-context for embodied agents

Next, we go from compatible object detections from human video, to interactible objects in embodied agents' environments. The compatibility score in Equation 3 is defined over the vocabulary of detectable objects $\mathcal{O}_V$, which may not be the same as the interactible objects $\mathcal{O}$ in the agent's environment. This mismatch is to be expected since we rely on in-the-wild video that is not carefully curated for the particular environments or tasks faced by the agents we train. To account for this, we first map objects in video and those in the agent's domain to a common Glove [45] word embedding space. Then, we re-estimate $\phi(o_m, o_n)$ for each object pair in $\mathcal{O}$ by considering nearest neighbors to $\mathcal{O}_V$ in this space. Specifically, in Equation 3, we set

$$S_v(o_m, o_n) = \sum_{o_i \in \mathcal{N}(o_m)} \sum_{o_j \in \mathcal{N}(o_n)} \sigma_{m,i} \sigma_{n,j} S_v(o_i, o_j), \qquad (4)$$

where $\mathcal{N}(o)$ are the nearest neighbors of environment object $o$ among video objects $\mathcal{O}_V$ within a fixed distance in embedding space, and $\sigma_{i,j}$ measures the dot product similarity of the corresponding word embeddings of objects $o_i$ and $o_j$. The resulting compatibility scores represent relationships between objects in the agent's environment, as inferred from similar objects in egocentric video. See Fig. 2 (right) for an illustration of top ACOs compared to other heuristics (like word embedding similarity) and Supp. for further details.

### 3.4 Learning to interact with an activity-context reward

Next, we cast the compatibility scores learned from video into an auxiliary reward for embodied agents. In short, the agent is rewarded for interactions with any objects at any time; however, this reward is enhanced when there are compatible ACOs nearby. For example, it is more valuable to turn-on the stove when there is a pan on it than if there was a knife (or no other objects) nearby. To maximize this reward, the agent must intelligently move objects in the environment to locations with compatible objects before performing interactions.

To achieve this, the agent maintains a persistent memory to keep track of where objects were moved to, and what other objects near it are potential ACOs (and thus, how it affects the value of nearby object interactions). The memory $\mathcal{M}$ starts empty for all objects. At each time-step,

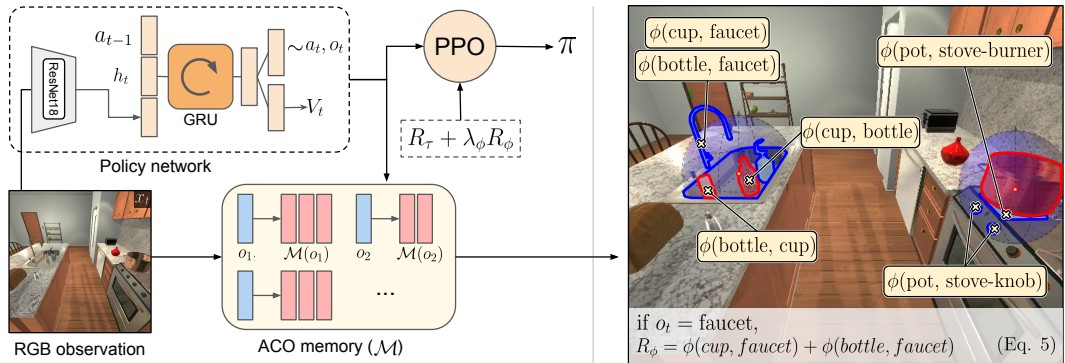

Figure 3: **Policy training framework**. **Left panel:** Our policy network generates an action $(a_t, o_t)$ given the current state observation. If objects are moved by the agent, the ACO memory $\mathcal{M}$ is updated to add objects (red) to their ACO's memory (blue) following Sec. 3.4. **Right panel:** An auxiliary reward $R_\phi$ is provided for object interactions based on nearby ACOs. For example, if the interaction target $o_t$ is the faucet, the agent is rewarded for having the cup and bottle nearby.

if an agent holding an object $o$ *puts-down* the object at location $p$ in 3D space[1], we identify $o$'s neighboring objects. For each neighboring object $o'$, $o$ is a potential ACO and is added to the memory: $\mathcal{M}(o') \bigcup \{(o, p) \mid d(o, o') < \epsilon\}$. We set $\epsilon = 0.5$m. Conversely, if an object $o$ is *picked-up* from a location $p$, it is removed from the list of potential ACOs of all objects it originally neighbored: $\mathcal{M}(o') \smallsetminus \{(o, p)\} \, \forall o' \in \mathcal{M}$, and its own ACO memory is cleared: $\mathcal{M}(o) = \phi$. Fig. 3 (right) shows a snapshot of this memory built until time-step $t-1$. Each object (in red) was placed at its respective locations in previous time-steps, and added to the ACO memories of nearby objects (in blue).

At time-step $t$ the agent is rewarded if it successfully interacts with an object $o_t$ based on this memory. The total activity-context reward $R_\phi$ is the sum of compatibility scores for objects in memory for which $o_t$ is a candidate ACO:

$$R_\phi(s_t, a_t, o_t, \mathcal{M}) = \begin{cases} \sum_{o' \in \mathcal{M}(o_t)} \phi(o', o_t) & \text{if } a_t \in \mathcal{A}_I \wedge c(a_t, o_t) = 0 \\ 0 & \text{otherwise,} \end{cases} \quad (5)$$

where $c(a_t, o_t)$ counts the number of times a particular interaction occurs. Note that the currently held object (or *null*, if nothing is held) is included in $\mathcal{M}$ for all interactions. See Supp. for pseudo-code of the memory update and reward allocation step.

Unlike the sparse task reward that is offered only at the goal state (Sec. 3.1), the activity-context reward is provided at every time-step. These dense rewards are not task-specific, meaning we do not give any privileged information about which video priors are most beneficial to a given task $\tau$. However, they nonetheless encourage the agent to take meaningful actions towards activities, helping to reach states that are strongly aligned with common task subgoals in human videos.

### 3.5 Interaction policy training

Putting it together, our training process is as follows. We adopt an actor-critic policy to train our agent [60]. At each time-step $t$, the current egocentric frame $x_t$ is encoded using a ResNet-18 [29] encoder and then average pooled and fed to an LSTM based recurrent neural network (along with encodings of the held object class and previous action) to aggregate observations over time, and finally to an actor-critic network (MLP) to generate the next action distribution and value. See Fig. 3 (left). In parallel, the agent maintains and updates its activity-context memory and generates an auxiliary reward based on nearby context objects at every time-step following Sec. 3.4.

The final reward is a combination of the task reward (Sec. 3.1) and the activity-context reward:

$$R(s_t, a_t, o_t, s_{t+1}, \mathcal{M}) = R_\tau(s_t, a_t, o_t, s_{t+1}) + \lambda_\phi R_\phi(s_t, a_t, o_t, \mathcal{M}), \quad (6)$$

where $\lambda_\phi$ controls the contribution of the auxiliary reward term. We train our agents using DD-PPO [64] for 5M steps, with rollouts of $T = 256$ time steps. Our model and all baselines use visual

---

[1]We calculate $p$ from the agent's pose and depth observations using an inverse projection transformation on the interaction target point in the agent's current view.

encoders from agents that are pre-trained for interaction exploration [40] for 5M steps, which we find benefits all approaches. See Fig. 3 and Supp. for architecture, hyperparameter and training details.

## 4 Experiments

We evaluate how well our agents learn complex interaction tasks using our human video based reward.

**Simulator and video datasets.** To train policies, we use the AI2-iTHOR [33] simulator where agents can navigate: $\mathcal{A}_N$ = {move forward, turn left/right 90°, look up/down 30°}, and interact with objects: $\mathcal{A}_I$ = {take, put, open, close, toggle-on, toggle-off, slice}. Our action space of size $|\mathcal{A}| = 110$ is the union of all navigation actions and valid object interactions following [63]. We use all 30 kitchen scenes from AI2-iTHOR, split into training (25) and testing (5) sets. To learn activity-context priors, we use all 55 hours of video from EPIC-Kitchens [13], which contains egocentric videos of daily, unscripted kitchen activities in a variety of homes. It consists of ~40k video clips annotated for interactions spanning 352 objects ($\mathcal{O}_V$) and 125 actions. Note that we use clip boundaries to segment actions, but we do not use the action labels in our method.

Since kitchen scenes present a diverse set of object interactions in multi-step cooking activities, they are of great interest in this research domain [13, 23, 40]. Further, the alignment of domain with AI2-iTHOR's kitchen environments provides a path to transfer knowledge from videos to agents.

**Visual semantic planning tasks.** We consider seven tasks in our experiments where the agent must **STORE**: put an object that is outside into a drawer, **HEAT**: turn on the stove with a cooking receptacle on it, **COOL**: store a food item or container inside the fridge, **CLEAN**: put an object in the sink and turn on the faucet to wash it, **SLICE**: slice a food item with a knife, **PREP**: place a food item inside a pot/pan, **TRASH**: throw an object into the trash bin,

Each task is associated with a goal state that the environment must be in. For example in the COOL task, an object that was originally outside, must be inside the fridge, and the fridge door must be closed. See Supp. for goal states for all tasks. These tasks represent realistic scenarios in home robot assistant settings, consistent with tasks studied in recent work [59, 40].

We generate 64 episodes per task and per environment with randomized object and agent positions to evaluate our agents. We report task success rates (%) on unseen test environments. This means that our model must both generalize what it observes in the real-world human video to the agent's egocentric views, as well as generalize from training environments to novel test environments.

**Baselines.** We compare several methods:

- **VANILLA** trains a policy using only the task reward (Sec. 3.1). This is the standard approach to train reinforcement learning agents.
- **SCENEPRIORS** [66] also uses only the task reward, but encodes spatial co-occurrences between objects using a graph convolutional network (GCN) model to enhance its state representation. We use the authors' code.
- **NAVEXP** [50] adds an auxiliary navigation exploration reward to the vanilla model. We use object coverage [17, 50], which rewards the agent for visiting (but not interacting with) new objects.
- **INTEXP** [40] is a state-of-the-art exploration method for object interaction that adds a reward for successful object interactions regardless of nearby context objects. We use the authors' code.

SCENEPRIORS represents the conventional approach of introducing visual priors into policy learning—through GCN encoders, not auxiliary rewards—and considers spatial co-occurrence instead of activity based priors. NAVEXP and INTEXP are exploration methods from recent work that incentivize all locations or objects equally. In contrast, our reward is a function of the state of the environment, and how well it aligns with observed states in video of human activity.

### 4.1 Visual semantic planning results

Table 1 shows success rates across all tasks. The numbers in brackets represent the probability that an agent executing random actions will reach the goal state assuming all objects are within reach. *Interaction-heavy* tasks requiring long sequences of steps like STORE (pick up object, open drawer, put object inside, close drawer) are significantly harder than *interaction-light* tasks like PREP (pick up food item, put in pan)—in this particular example, by four orders of magnitude.

Our approach outperforms prior work on all interaction-heavy tasks, and performs competitively on interaction-light tasks. Compared to the VANILLA baseline, SCENEPRIORS's stronger observation

| | COOL(1e-7) | STORE(1e-7) | HEAT(2e-6) | CLEAN(2e-5) | SLICE(1e-3) | PREP(2e-3) | TRASH(2e-3) |
|---|---|---|---|---|---|---|---|
| VANILLA | 0.12±0.07 | 0.00±0.00 | 0.01±0.01 | 0.35±0.12 | 0.30±0.03 | 0.22±0.05 | 0.14±0.05 |
| SCENEPRIORS [66] | 0.14±0.09 | 0.00±0.00 | 0.04±0.01 | 0.35±0.02 | **0.36**±0.05 | 0.26±0.08 | 0.20±0.06 |
| NAVEXP [50] | 0.05±0.04 | 0.01±0.01 | 0.01±0.01 | 0.43±0.02 | 0.29±0.05 | **0.33**±0.02 | **0.25**±0.04 |
| INTEXP [40] | 0.11±0.06 | 0.03±0.03 | 0.06±0.03 | 0.19±0.05 | 0.26±0.07 | 0.19±0.08 | 0.02±0.01 |
| OURS | **0.26**±0.06 | **0.12**±0.07 | **0.13**±0.05 | **0.53**±0.03 | **0.36**±0.06 | 0.26±0.07 | 0.13±0.02 |

Table 1: **Task success rates (%) on test environments.** Numbers in brackets indicate the probability that a random agent performs the correct sequence of interactions to complete a task (e.g., COOL is much harder than PREP). Our video-based activity-context priors result in the best performance across all interaction-heavy tasks. Navigation-based exploration agents perform well for tasks that involve minimal object interactions (e.g. PREP, TRASH). Values are averaged over 3 training runs.

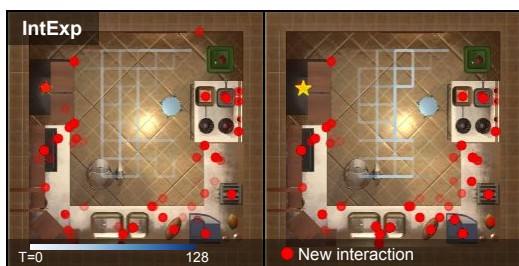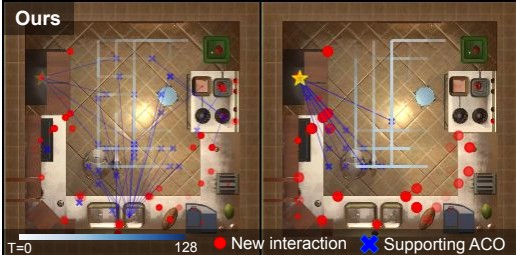

Figure 4: **Policy rollouts for the COOL task during early/late stage training**. Red dots represent auxiliary rewards (size represents value). Our agents learn to move objects to suitable locations (near supporting ACOs, blue crosses) to increase their reward, rather than simply visit *more* objects. See text.

encoding results in better performance with the same task reward; however, while its spatial co-occurrence prior encoding what objects are *found together* (rather than *used together*) is helpful for object search [66], it falls short for object interactions.

INTEXP provides a task-agnostic objective for pre-training interaction agents; it results in strong visual encoder features, but yields poor performance when used as an auxiliary reward. It fails to discriminate between useful interaction sequences that are aligned with task goals (e.g., put pan on stove, turn-on stove) and arbitrary interaction sequences that maximize its reward (e.g., pick up, then put down objects sequentially). NAVEXP performs poorly on interaction-heavy tasks, often achieving close to zero success rates. However, it performs well on tasks like TRASH and PREP where a single object must be moved to a target location for success.

Fig. 4 shows qualitative results of 100 policy rollouts. Each panel shows a side-by-side comparison of agent behavior during early (1M steps) and late (5M steps) training for the same task. Each red dot represents an auxiliary reward given to the agent, sized by reward value; the yellow star denotes where the task reward is issued. INTEXP (left panel) is rewarded uniformly for each interaction (equally sized red dots), and thus has a single strategy to maximize reward — perform *more* interactions. In contrast, our approach (right panel) can maximize reward along two axes: perform more interactions as before, or selectively move objects to favorable positions near ACOs (blue crosses). This guides agent exploration to relevant states for activities, translating to higher task success.

## 4.2 Training speed and task success versus difficulty

Fig. 5 shows consolidated results across all tasks, treating each episode of each task as an individual instance that can be successful or not. These results give a sense of overall performance of each baseline, despite variation across tasks. See Supp. for a task-specific breakdown of results.

Fig. 5 (left) shows convergence speed. Our approach sees an increased success rate at early training epochs as our dense rewards allow agents to make progress towards goals without explicit task rewards. NAVEXP performs poorly in early iterations as it prioritizes navigation actions to visit as many objects as possible. Overall this result shows a key advantage of our idea: learning from human video not only boosts overall policy success rates, but it also accelerates the agent's learning process, giving a "head start" from having observed how people perform general daily interactions. Importantly, our ablations (below) show that access to the video alone is not sufficient; our idea to capture functional activity-contexts is key.

Fig. 5 (right) analyzes task success as a function of navigation difficulty. We measure this as the ideal geodesic distance to each of the objects required to reach the goal state. This value is low when

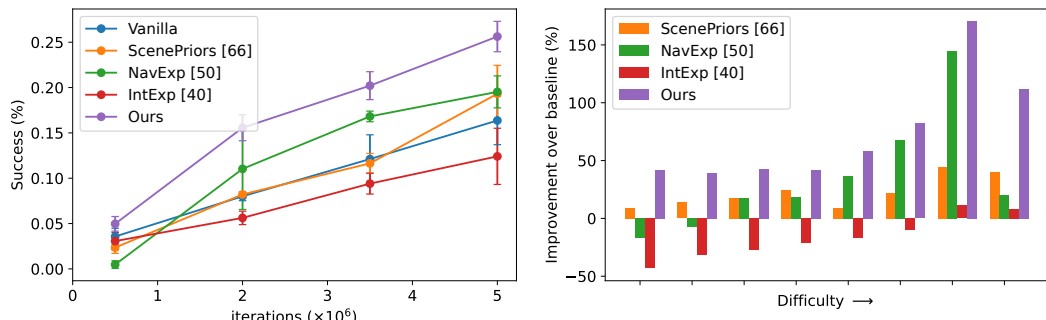

Figure 5: **Consolidated performance across all seven tasks. Left: Success rate vs. training iteration.** Our dense activity-context rewards accelerate performance at early training iterations. **Right: Success rate improvement vs. navigation difficulty.** Episodes are sorted and grouped by ideal navigation distance to task objects (low → high). Our method shows largest improvements over VANILLA on difficult instances.

|  | COOL | STORE | HEAT | CLEAN | SLICE | PREP | TRASH |
|---|---|---|---|---|---|---|---|
| UNIFORM | $0.07_{\pm 0.00}$ | $0.02_{\pm 0.01}$ | $0.12_{\pm 0.00}$ | $0.37_{\pm 0.05}$ | $0.26_{\pm 0.03}$ | $0.22_{\pm 0.05}$ | $0.02_{\pm 0.01}$ |
| WORDEMBED | $0.19_{\pm 0.03}$ | $0.02_{\pm 0.02}$ | $0.03_{\pm 0.00}$ | $0.40_{\pm 0.13}$ | $0.34_{\pm 0.01}$ | $0.22_{\pm 0.01}$ | $0.04_{\pm 0.02}$ |
| SPATIALCOOC | $\mathbf{0.33}_{\pm 0.05}$ | $0.02_{\pm 0.01}$ | $0.06_{\pm 0.02}$ | $0.47_{\pm 0.09}$ | $0.33_{\pm 0.02}$ | $0.25_{\pm 0.08}$ | $\mathbf{0.13}_{\pm 0.05}$ |
| OURS (INTSEQ) | $\mathbf{0.33}_{\pm 0.04}$ | $0.11_{\pm 0.03}$ | $0.07_{\pm 0.03}$ | $0.43_{\pm 0.04}$ | $0.30_{\pm 0.05}$ | $0.20_{\pm 0.02}$ | $0.09_{\pm 0.04}$ |
| OURS (ACO) | $0.26_{\pm 0.06}$ | $\mathbf{0.12}_{\pm 0.07}$ | $\mathbf{0.13}_{\pm 0.05}$ | $\mathbf{0.53}_{\pm 0.03}$ | $\mathbf{0.36}_{\pm 0.06}$ | $\mathbf{0.26}_{\pm 0.07}$ | $\mathbf{0.13}_{\pm 0.02}$ |

Table 2: **Compatibility score ablations.** Simple similarity measures (WORDEMBED) or naively using interaction labels in video (INTSEQ) produces sub-optimal policies. Our activity-context detection based priors perform the best across the majority of tasks. Results are averaged over 3 runs.

objects are close by in small environments, and large when the agent has to move around to find specific objects in large environments. We sort episodes by this criterion into 8 groups of increasing difficulty and compare each method against the VANILLA baseline policy. Our method has the highest success rates across difficulties and offers the largest improvements on more difficult instances.

### 4.3   Compatibility function ablations

Finally we investigate how different sources of priors impact our agents when incorporated into our reward scheme. Specifically, we vary how we compute our compatibility score $\phi(o_i, o_j)$ in Equation 3. **UNIFORM** assigns equal compatibility to all object pairs; **WORDEMBED** uses Glove [45] embedding similarity; **SPATIALCOOC** uses spatial co-occurrence statistics derived from static images[2]; **OURS (INTSEQ)** uses transition probabilities between object interactions from ground truth, labeled sequences of the same egocentric videos.

Table 2 shows the results. Uniform or semantic similarity heuristics are insufficient to capture the role of objects participating in activities. SPATIALCOOC's object co-occurrences are aligned for some tasks (e.g., vegetables are co-located with fridges in static images, and activities involving storing groceries), but also capture unhelpful static relationships (see Fig. 2, right). INTSEQ benefits from egocentric video; however, it uses ground truth video object labels, and falsely assumes that the next object interacted with in time is necessarily relevant to the previous object. Our activity-context score is tightly linked to each object's involvement in interactions, and proves to be the most useful prior.

## 5   Conclusion

We proposed an approach to discover activity-context priors from real-world human egocentric video and use them to accelerate training of interaction agents. Our work provides an avenue for embodied agents to directly benefit from human experience even when the tasks and environments differ. Future work could explore policies that update human-activity prior beliefs during task-specific policy training, and policy architectures that encode such priors jointly in the state representation.

**Acknowledgments**: Thanks to Santhosh Ramakrishnan for helpful feedback and the UT Systems Admin team for their continued support. UT Austin is supported in part by DARPA L2M, Facebook Reality Labs, and the UT Austin NSF AI Institute. K.G. is paid as a Research Scientist by Facebook AI.

---

[2]This is different from SCENEPRIORS (Sec. 4.1) which encodes the same priors in the state representation.

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
