## Supplementary Material

This section contains supplementary material to support the main paper text. The contents include:

- (§S1) A video demonstrating our agents ACO memory creation process and reward calculation described in Sec. 3.4.
- (§S2) A video comparing our agents to baseline policies to supplement Fig. 4.
- (§S3) Algorithm for creating and updating ACO memory described in Sec. 3.4.
- (§S4) Figures illustrating the correspondences between scenes in EPIC Kitchens and observations in THOR using our method (Sec. 3.3)
- (§S5) Additional qualitative figures to supplement Fig. 2 in the main paper.
- (§S6) Detection model architecture details and additional implementation details related to Sec. 3.2
- (§S7) Policy architecture details, training details and hyperparameters for our model in Sec. 3.5
- (§S8) Implementation details for baselines in Sec. 4 (Baselines).
- (§S9) Goal descriptions for each task presented in Sec. 4 (Tasks).
- (§S11) Task-specific breakdown of results presented in Sec. 4.1 and Sec. 4.2.
- (§S10) Additional experiments varying model architecture and combined reward schemes to supplement Sec. 4.

## S1    ACO memory creation and reward calculation demo video

In the video, we show the process of creating and building the activity-context memory described in Sec. 3.4. The video shows the first-person view of the agent as it performs a series of object interactions (top left). We overlay masks for objects that are manipulated (red), nearby objects whose activity-context object (ACO) memory is updated (blue), the distance threshold under which objects are considered *near* to each other (teal). Finally, we show how the presence of these objects affects the reward provided to the agent (top right). The video illustrates that bringing a mug to the sink or a pot to the stove results in high rewards as they are aligned with activities, while bringing a knife to the garbage bin is far less valuable.

## S2    Video demo of our policy compared to baselines

In the video, we show rollouts of various policies during training to supplement Fig. 4 in the main paper. The video compares the baseline approaches to our method. It shows each step in a trajectory and the corresponding reward given to the agent, illustrating the contribution of activity-context objects to the total reward in our method compared to the uniform reward provided in the baselines. As mentioned in Sec. 4.1, our agents receive variable rewards (non-uniformly sized red dots) based on nearby ACOs (blue crosses), which encourages agents to move objects to more meaningful positions aligned with activities.

## S3    ACO memory creation and update algorithm

We present the algorithm for creating and maintaining the ACO memory in Algorithm 1. This corresponds to the steps outlined in Sec. 3.4 of the main paper and the accompanying reward equation Equation 5. Note that in practice, we normalize $\phi(o_t, o)$ in L16 of Algorithm 1 such that the maximum rewarding ACO offers a reward of 1.0, to ensure that agents do not ignore objects with scores distributed across many potential ACOs.

## S4    ACO correspondences between EPIC and THOR scenes

As mentioned in Sec. 3.3 we translate from ACO pairs learned in video to a reward used in simulation. We show qualitative results for which scenes in THOR correspond to activities observed in EPIC in Fig. S1. The first column of each row shows a frame from an EPIC video clip showing a particular human activity. The remaining columns show similar "states" from THOR that our agents deem

**Algorithm 1** Activity-context reward memory.

**Input:** ACO memory $\mathcal{M}$, visitation count $c$, distance metric $d$, distance threshold $\epsilon$
**Input:** State $s_t$, action $(a_t, o_t)$, held object $o$ at time-step $t$
 1: **function** UPDATEMEM($s_t, a_t, o_t, \mathcal{M}$)
 2:     **if** $a_t = $ "put" **then**                                                                           ▷ Put $o$ at position $p$
 3:         $\mathcal{M}(o') \leftarrow \mathcal{M}(o') \bigcup \{(o, p) \mid d(o, o') < \epsilon\}$
 4:     **else if** $a_t = $ "take" **then**                                                             ▷ Take $o_t$ from location $p_t$
 5:         $\mathcal{M}(o_t) \leftarrow \{\}$
 6:         $\mathcal{M}(o') \leftarrow \mathcal{M}(o') \smallsetminus \{(o_t, p_t)\} \mid \forall o' \in \mathcal{M}$
 7:     **end if**
 8:     **return** Updated memory $\mathcal{M}$
 9: **end function**
10:
11: **function** ACREWARD($s_t, a_t, o_t, \mathcal{M}$)
12:     **if** $a_t \notin \mathcal{A}_\mathcal{I}$ **or** $c(a_t, o_t) > 0$ **then return** $0$ ;
13:     $\mathcal{M} \leftarrow$ UPDATEMEM($s_t, a_t, o_t, \mathcal{M},$)
14:     $Z \leftarrow \max\limits_{o \in \mathcal{O}} \phi(o_t, o)$
15:     $R_{ACO} \leftarrow \sum\limits_{o \in \mathcal{M}(o_t)} \phi(o_t, o)/Z$                                      ▷ Equation 5
16:     $c(a_t, o_t) \leftarrow c(a_t, o_t) + 1$
17:     **return** $R_\phi$
18: **end function**

desirable to reach based on the distribution of ACOs present. Interactions performed in these states are highly rewarded following our approach.

The figures also illustrate our automatic mapping from the video vocabulary $\mathcal{O}_V$ to the agent environment vocabulary $\mathcal{O}$ for objects using word embedding similarity described in Sec. 3.3. For example, "garlic:paste" in EPIC is mapped to other food-like objects in THOR like "tomatoes" (left, bottom row); "drawers" are mapped to both "cabinets" and "drawers" (right, top row).

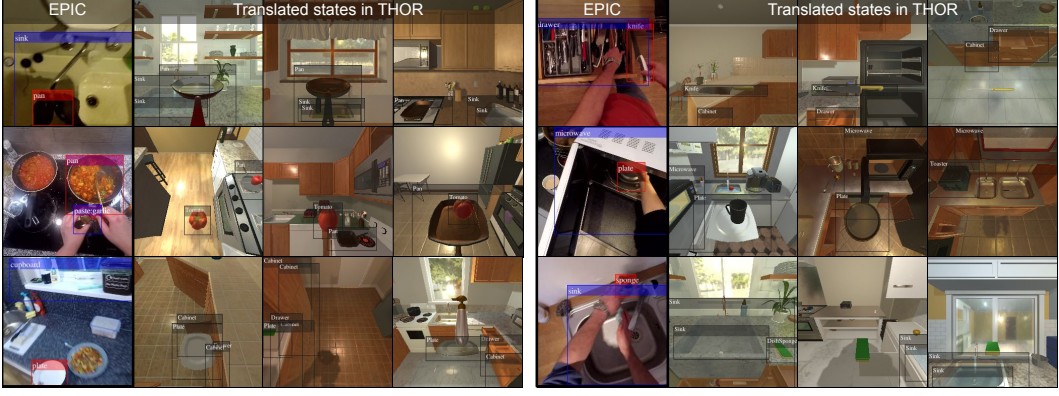

Figure S1: **Discovered EPIC $\longleftrightarrow$ THOR ACO correspondences.** First column shows a human activity from EPIC. Subsequent columns show similar states from THOR which provide high rewards when interactions are performed with objects once the agent is in that state.

## S5  Additional ACO detection results on EPIC-Kitchens

We show additional detection results to supplement Fig. 2 (left). These images show sampled (object, ACO) tuples (Equation 2) in red and blue respectively. The last column shows failure cases due to incorrect active object detections and incorrect object instance detection.

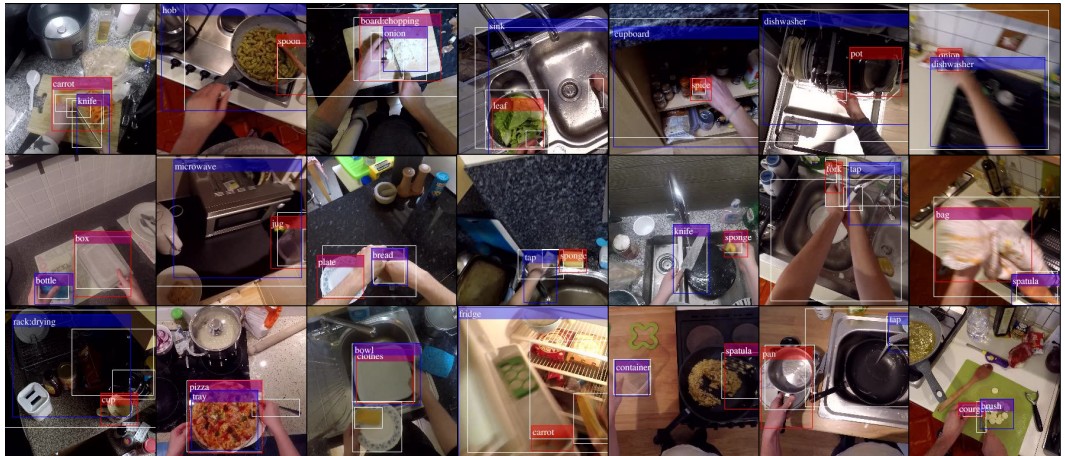

Figure S2: **Additional EPIC detections to supplement Fig. 2.** Last column shows failure cases.

## S6   Pre-trained detection model and ACO scoring details

**Pre-trained detection models**   As mentioned in Sec. 3.2, we use two detection models in our approach – (1) An active object detector which generates high-confidence box proposals for objects being interacted with hands (but does not assign object class labels to them) and (2) An object instance detector that produces a set of named objects and boxes for visible object instances. For (1) we use pre-trained models provided by authors of [56][3]. For (2) we use pre-computed detections per frame using a Faster-RCNN model released by the authors of EPIC-Kitchens [13]. We set confidence thresholds of 0.5 for all models.

**Activity context curation details**   To infer each frame's activity-context following Equation 2, we use a manually curated list of *moveable* objects from EPIC, though it is possible to automatically infer this list from action labels on video clips (all objects that are *picked* up), or using the aforementioned hand-object detectors. Of the 398 objects in EPIC, 349 are moveable. We list the remaining objects in the table below.

| tap | top | microwave | machine:washing | toaster | machine:sous:vide | flame |
|-----|-----|-----------|-----------------|---------|-------------------|-------|
| cupboard | oven | button | processor:food | knob | window | fire |
| drawer | maker:coffee | juicer | plug | ladder | heater | grill |
| hand | sink | scale | kitchen | wall | door:kitchen | time |
| fridge | heat | rack:drying | floor | tv | table | timer |
| hob | dishwasher | freezer | fan:extractor | shelf | rug | desk |
| bin | blender | light | chair | stand | switch | lamp |

**ACO mapping details**   As mentioned in Sec. 3.3, to match object classes between AI2-iTHOR and EPIC Kitchens we map each (object, ACO) tuple in the video object space $\mathcal{O}_V$ to corresponding tuples in environment object space $\mathcal{O}$ following Equation 4. We use a GloVe [45] similarity threshold of 0.6. Lower values lead to undesirable mappings across object classes (e.g., toasters and refrigerators which are both appliances, but participate in distinct activities) .

## S7   Policy architecture and training details

We provide additional architecture and training details to supplement information provided in Sec. 3.5 in the main paper.

**Policy network**   As mentioned in Sec. 3.5, we use a ResNet-18 observation encoder pretrained with observations from 5M iterations of training of an interaction exploration agent [40]. We transfer

---

[3]https://github.com/ddshan/hand_object_detector

the backbone only and freeze it during task training. Each RGB observation is encoded to a 512-D feature. A single linear embedding layer is used to embed the previous action and the currently held object (or *null*) to vectors of dimension 32 each. The total observation feature is the concatenation of these three features. All architecture hyperparams are listed in Table S1 (Policy network).

**Training hyperparameters**    We modify the Habitat-Lab codebase [52] to support training agents in the THOR simulator platform [33]. We search over $\lambda_\phi \in \{0.01, 0.1, 1.0, 5.0\}$ for Equation 6 and select $\lambda_\phi = 1.0$ which has the highest consolidated performance on validation episodes for all methods following the procedure in Sec. 4.2. All training hyperparameters are listed in Table S1 (RL training).

| RL training | |
| --- | --- |
| Optimizer | Adam |
| Learning rate | 2.5e-4 |
| # parallel actors | 64 |
| PPO mini-batches | 2 |
| PPO epochs | 4 |
| PPO clip param | 0.2 |
| Value loss coefficient | 0.5 |
| Entropy coefficient | 0.01 |
| Normalized advantage? | Yes |
| Training episode length | 256 |
| LSTM history length | 256 |
| # training steps ($\times$ 1e6) | 5 |
| Policy network | |
| Backbone | resnet18 |
| Input image size | 256$\times$256 |
| LSTM hidden size | 512 |
| # layers | 2 |

Table S1: **RL policy architecture and training hyperparameters.**

## S8    Baseline implementation details

We present implementation details for baselines in Sec. 4 (Baselines). NAVEXP and INTEXP baselines use the same architecture as our model described in Sec. S7, but vary in the reward they receive during training. SCENEPRIORS uses a different backbone architecture that uses a GCN based state encoder as described below.

**NAVEXP**    Agents are rewarded for visiting new objects such that the object is visible and within interaction range (less than $1.5m$ away). A constant reward is provided for every new object class visited. This is similar to previous implementations [50, 17, 40]. We use the implementation from [40].

**INTEXP**    Following [40], agents are rewarded for new object interactions. The reward provided has the form in Equation 5, but provides a constant reward regardless of ACOs present. We use the author's code.

**SCENEPRIORS**    We modify the architecture in [66], which was built for object search. We use the author's code. First, we remove the goal object encoding as the agent is not searching for a single object. Second, we replace the backbone network with our shared ResNet backbone to ensure fair comparison. We use a GCN encoding dimension of $512$. The remaining architecture details are consistent with [66].

## S9   Goal condition details for all tasks

We next list out formal goal conditions for all tasks described in Sec. 4 of the main paper (Tasks). Each goal is specified as a conjunction of predicates that need to be satisfied in a candidate goal state. The goal is satisfied if for any object $o$ in the environment, the following conditions are true:

- STORE: inReceptacle(o, Drawer) $\land$ isClosed(Drawer) $\land$ isStorable(o)
- HEAT: inReceptacle(o, StoveBurner) $\land$ isToggledOn(StoveKnob) $\land$ isHeatable(o)
- COOL: inReceptacle(o, Fridge) $\land$ isClosed(Fridge) $\land$ isCoolable(o)
- CLEAN: inReceptacle(o, SinkBasin) $\land$ isToggledOn(Faucet) $\land$ isCleanable(o)
- SLICE: isHolding(Knife) $\land$ isSliceable(o)
- PREP: [inReceptacle(o, Pot) | inReceptacle(o, Pan)] $\land$ isCookable(o)
- TRASH: inReceptacle(o, GarbageCan) $\land$ isTrashable(o)

where *inReceptacle* checks if an object is inside/on top of a particular object, *is-X-able* filters for objects with specific affordances (e.g., only objects that can be placed on the stove like pots/pans/kettles satisfy *isHeatable*), *isClosed* and *isToggledOn* checks for specific object states, and *isHolding* checks if the agent is holding a specific type of object (e.g., for SLICE this has to be a Knife or a ButterKnife). Further, for each task that involves moving objects to receptacles, the object must originally have been outside the receptacle (e.g., outside the Fridge for COOL; off the stovetop for HEAT).

## S10   Additional ablation experiments

We present a comparison of different backbone architectures (ResNet18 vs. ResNet50) and aggregation modules (LSTM vs. GRU) for both our model and the baselines in Table S3. We evaluate on the unseen test episodes for 4 interaction-heavy tasks. The average results of 2 training runs are in the table below. Using stronger backbones seems to help marginally, but does not offer conclusive results. Using the simpler GRU based aggregation (instead of LSTM) results in large improvements. Overall, the trends remain consistent across all configurations: Vanilla < NavExp < Ours. Architectural changes alone in the baselines (to either the backbone, or the aggregation mechanism) are not enough to compensate for task difficulty — performance on Cool, Store and Heat remain low (<10%) for Vanilla and NavExp.

In Table S2 we show the results of our policies combined with rewards from a navigation exploration agent. As mentioned in Sec. 4.1, while our agent excels in interaction-heavy tasks, navigation exploration agents perform well on interaction-light tasks which often require finding a single object and bringing it to the right location. For example in the Trash task, there is a single garbage can that navigation agents quickly find as they cover area, but that our agents struggle to find early on. The two strategies can be combined to address this issue. Table S2 shows average results of 2 runs where we add the two reward functions together with equal weights (=0.5) to achieve the best performance.

| | COOL | STORE | HEAT | CLEAN | SLICE | PREP | TRASH |
|---|---|---|---|---|---|---|---|
| NAVEXP [50] | 0.05 | 0.01 | 0.01 | 0.43 | 0.29 | 0.33 | 0.25 |
| OURS | **0.26** | **0.12** | 0.13 | **0.53** | **0.36** | 0.26 | 0.13 |
| OURS + NAVEXP | 0.25 | 0.05 | **0.19** | 0.50 | 0.34 | **0.41** | **0.26** |

Table S2: **Combined policy experiments.** Navigation exploration offers complementary reward signals that can be combined with our method for stronger performance.

| | ResNet18 + LSTM | | | | ResNet50 + LSTM | | | | ResNet18 + GRU | | | |
|---|---|---|---|---|---|---|---|---|---|---|---|---|
| | COOL | STORE | HEAT | CLEAN | COOL | STORE | HEAT | CLEAN | COOL | STORE | HEAT | CLEAN |
| VANILLA | 0.07 | 0.00 | 0.02 | 0.29 | 0.04 | 0.00 | 0.01 | 0.26 | 0.31 | 0.03 | 0.03 | 0.38 |
| NAVEXP [50] | 0.02 | 0.02 | 0.01 | 0.44 | 0.02 | 0.00 | 0.00 | **0.42** | 0.00 | 0.00 | 0.03 | 0.35 |
| OURS | **0.30** | **0.16** | **0.11** | **0.55** | **0.15** | **0.11** | **0.12** | 0.36 | **0.52** | **0.31** | **0.17** | **0.73** |

Table S3: **Backbone architecture ablations.** Performance on four interaction-heavy tasks.

## S11    Task-specific results breakdown

We include task level breakdowns of results from Sec. 4.2 of the main paper. These results highlight the strengths and weaknesses of all models on a task-specific level to supplement overall task success results in Table 1 of the main paper. These include Task success vs. training iteration (Fig. S3) and Task success vs. instance difficulty (Fig. S4)

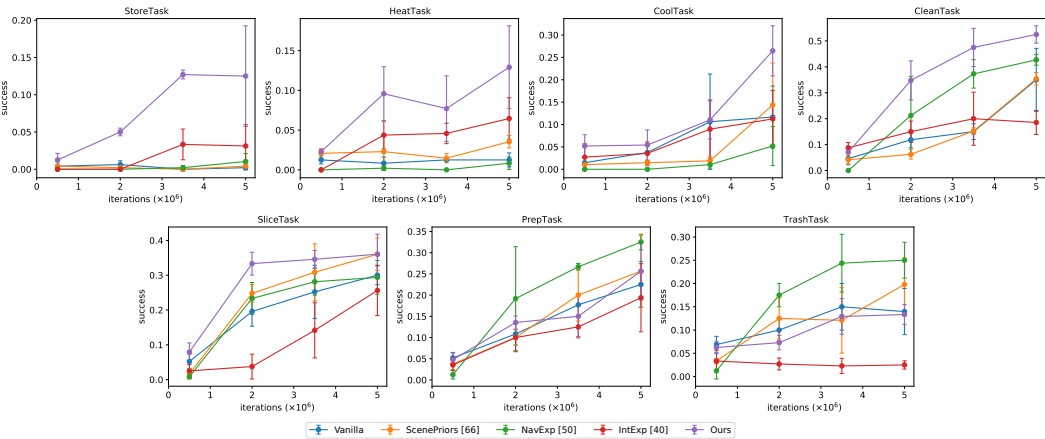

Figure S3: **Task success vs. training iteration.** This is the task-specific version of Fig. 5 (left) that shows convergence rates of all methods.

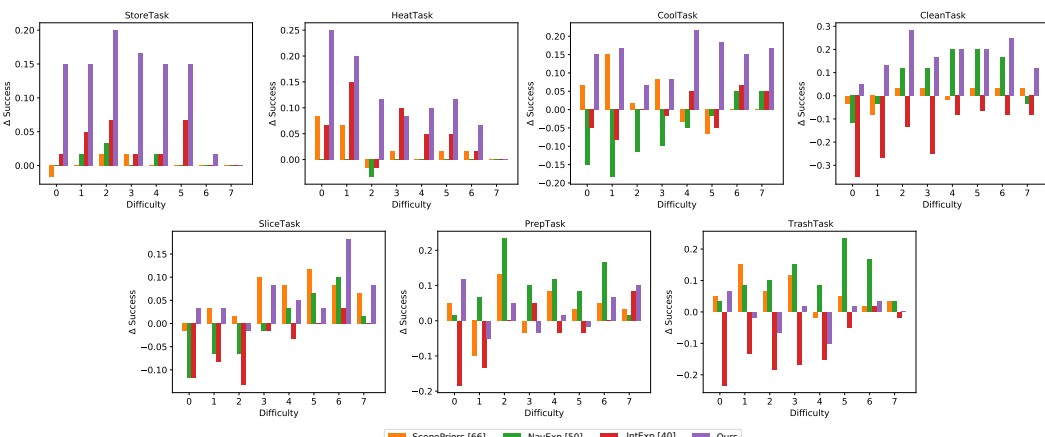

Figure S4: **Task success vs. navigation difficulty.** This is the task-specific version of Fig. 5 (right) that shows improvement in success over the baseline model. Note: we show absolute improvement (instead of relative) as the baseline has zero success for some tasks and some difficulty levels.