# OpenReview forum: "Shaping embodied agent behavior with activity-context priors from egocentric video"
_NeurIPS.cc/2021/Conference — NeurIPS 2021 Spotlight_

### Official Review · Reviewer_zAh2 · 2021-07-14

**Rating:** 7
**Confidence:** 3

**Summary:**

The paper introduces activity-context priors learned from real-world egocentric video frames, which are used as an auxiliary reward to encourage RL agents to solve longer-term object interaction tasks. Instead of directly imitating human videos, the prior is computed automatically using statistics from an off-the-shelf object recognition model run on the EPIC-Kitchens dataset.

**Limitations And Societal Impact:**

The authors do well in discussing limitations of their method, including the ACO detection errors and cases where simpler methods seem to perform better. One significant limitation is that the method relies on a hand-object interaction detection model, which restricts the possible sources of data which it can use for learning (other datasets than epic kitchens).

**Main Review:**

Originality: Although there are other methods which try to compute exploration rewards based on object interaction and novelty, the authors clearly explain how their method is novel. Compared to prior methods, this method specifically tries to encode not just which objects occur near each other, but which objects are actually used by the agent to interact together, which can reduce the impact of spurious correspondences. The experimental comparison is very thorough and the methods which the authors choose to compare to are very relevant.

Quality: The evaluation tasks are very challenging for RL and the method makes significant improvements on these difficult multi-step tasks.

The compatibility score ablations are thorough and insightful. The method is able to accelerate early training performance, when the algorithm still needs to rely on the dense provided rewards for exploration.

Clarity: The paper is very well written. Code will be released at a later date, but the appendix provides sufficient details for reproducibility at the moment. Addressing the following points may make things even better:

It took me a bit to figure out whether $\phi(o_i, o_j) == \phi(o_j, o_i)$, but I see that it is not. It could be helpful to mention this explicitly.

How sensitive is the method to the auxiliary reward scaling hyperparameter $\lambda_\phi$?

The intro states “Specifically, we 1) train visual models to detect how humans prepare their environment for activities from egocentric video”, however, maybe this is not quite precise -- the scores seem to model which objects occur in activities together, which is not solely preparation?

The discussion of how the memory $\mathcal{M}$ is maintained is rather complex and difficult to understand as written. It could be helpful to give a more general description of what the memory is keeping track of and when rewards are given before introducing details.

Significance: The proposed method can use existing egocentric real-world video to learn the priors, and does not need any additional demonstration data. This is impactful in that it may allow methods to better use existing large-scale data to improve the learning of RL algorithms.

However, the performance on tasks with minimal object interactions, while competitive, is not quite as strong as prior work. This is rather surprising as it seems like the proposed method should prioritize object interactions which solve the task and still perform better than navigation-based methods which weigh all object interactions equally. Is there an explanation for why performance is not as strong on those tasks?

Other comments:

Figure 4 is challenging to parse -- it is difficult to see what the correct sequence of object movements to solve the task should be (and what we would like the corresponding red dots to look like).

Figure 2 is great!

Recommendation:

The idea of learning priors specifically to encode activities rather than encouraging any object interactions seems like a promising direction. Comparisons are made to state of the art methods and the method performs well on the most difficult tasks. The paper is well written, the work is positioned and explained well compared to relevant prior work, and the idea is presented clearly. The main negative I see is the performance on the simpler (less object interaction) tasks. So, I think this paper meets the threshold and recommend acceptance.

**Time Spent Reviewing:**

3

---

> ### Author Response · Authors · 2021-08-10
> **Response to Reviewer zAh2**
>
> Thank you for the detailed suggestions and feedback.
>
> ---
> > It took me a bit to figure out whether $\phi(o_i, o_j) == \phi(o_j, o_i)$, but I see that it is not. It could be helpful to mention this explicitly.
>
> That's correct — $\phi(o_i, o_j)$ represents the value that object $o_i$ brings to interactions with object $o_j$ when $o_i$ is brought near $o_j$, which may not be symmetric. For instance, if $o_i$ is an immovable object (e.g. fridge, table, sink), $\phi(o_i, o_j)$ is zero while $\phi(o_j, o_i)$ may not be (see Eq 2. and Sec. S6 in Supp). We will make this clear at the end of Sec. 3.2.
>
> ---
> > How sensitive is the method to the auxiliary reward scaling hyperparameter ?
>
> As mentioned in Sec. S7 (in Supp), we select $\lambda_\phi$ based on the aggregate performance on validation episodes for two interaction-heavy tasks (Cool, Store) and two interaction-light tasks (Slice, Prep). The table below shows the results of the experiment.
>
> | $\lambda_\phi \rightarrow$ | *0.1* | *1.0*    | *5.0* |
> |----------------------------|-------|----------|-------|
> | *NavExp*                   | 0.25  | **0.27** | 0.00  |
> | *IntExp*                   | 0.14  | **0.17** | 0.00  |
> | *Ours*                     | 0.18  | **0.29** | 0.04  |
>
> For all methods, $\lambda_\phi = 1.0$ results in the highest validation performance. NavExp is relatively less sensitive to this parameter. For all methods, very high values of the parameter result in policies that ignore the task goal reward to focus solely on the exploration reward, resulting in trivially low performance.
>
> ---
> > The intro states "Specifically, we 1) train visual models to detect how humans prepare their environment for activities from egocentric video", however, maybe this is not quite precise -- the scores seem to model which objects occur in activities together, which is not solely preparation?
>
> As mentioned in L175, the scores represent which objects are *brought together* during an activity. It excludes objects that incidentally co-occur during an activity, but are not actually used (L158-9). While the objects are detected from the activity frames (e.g. "plates" and "sink" in washing plates), the objects had to be transported to the right location first ("plates" to the "sink") — this is what we meant by the phrase "prepare the environment".
>
> ---
> > The discussion of how the memory  is maintained is rather complex and difficult to understand as written. It could be helpful to give a more general description of what the memory is keeping track of and when rewards are given before introducing details.
>
> Thanks for the suggestion. We will briefly mention the overall goal of the memory in the beginning of Sec 3.4. In short, the memory keeps track of where objects were last moved to by the agent, and what other objects are near it (and thus, how it affects the value of nearby object interactions).
>
> ---
> > The proposed method ... does not need any additional demonstration data. This is impactful ... However, the performance on tasks with minimal object interactions, while competitive, is not quite as strong as prior work.
>
> > The main negative I see is the performance on the simpler (less object interaction) tasks. So, I think this paper meets the threshold and recommend acceptance.
>
> As mentioned in L291-2, the interaction-light tasks often require finding a single object and bringing it to the right location. For example in the Trash task, there is a single garbage can that navigation agents quickly find as they cover area, but that our agents struggle to find early on. The two strategies can be combined to address this issue. The table below shows average results of 2 runs where we add the two reward functions together with equal weights (=0.5).
>
> |                 | *Cool* | *Store* | *Heat* | *Clean* | *Slice* | *Prep* | *Trash* |
> |-----------------|--------|---------|--------|---------|---------|--------|---------|
> | *NavExp*        | 0.05   | 0.01    | 0.01   | 0.43    | 0.29    | 0.33   | 0.25    |
> | *Ours*          | 0.26   | 0.12    | 0.13   | 0.53    | 0.36    | 0.26   | 0.13    |
> | *Ours + NavExp* | 0.25   | 0.05    | 0.19   | 0.50    | 0.34    | 0.41   | 0.26    |
>
> Our combined agent outperforms the navigation-only agent on every task, and has similar performance to our original agent on most interaction-heavy tasks (except Store). Critically, it recovers the lost performance on the interaction-light tasks (Prep, Trash). Careful reward scaling may further improve performance.
>
> ---
> > Figure 4 is challenging to parse -- it is difficult to see what the correct sequence of object movements to solve the task should be (and what we would like the corresponding red dots to look like).
>
> The figure compares two methods to highlight: (1) the training signals available to agents and (2) how agent behavior changes over time as a result. IntExp agents receive uniform rewards for new interactions, regardless of how well aligned they are with human goals. As a result, there is only one avenue to increase rewards — perform more interactions. In contrast, our agents are rewarded non-uniformly depending on relevant ACOs (blue lines), and can thus increase their reward in two ways — perform more interactions as before, but also transport objects intelligently to enhance the value of the interactions (larger dot size). Optimizing for this accelerates training of our agents. We will make this clearer in L293-9.
>
> ---
> > One significant limitation is that the method relies on a hand-object interaction detection model, which restricts the possible sources of data which it can use for learning (other datasets than epic kitchens).
>
> Our approach is designed to transfer information about human-object interactions from video to simulation (L245-6). Video datasets that do not contain such human activity (e.g., third person video datasets like Kinetics) are therefore not relevant for our approach. Given our requirement, the use of a hand-object interaction detector does not further restrict the sources of data — we expect that other datasets with object interaction also contain detectable hands. Other popular video datasets that fit this description include EGTEA+ [1], HowTo100M [2], VLOG [3].
>
> More generally, our dataset should (1) have videos from similar domains involving similar objects (in our case, kitchen scenes) so that the compatibility scores (Eq. 3) are meaningful, and (2) capture goal driven human behavior and not, for example, an agent moving objects around randomly.  Overall, the alignment assumed is still fairly weak — it does not require matching embodiment or perfect visual fidelity, and is less restrictive than collecting task demonstrations as discussed in L37-9.
>
> - [1] In the eye of beholder: Joint learning of gaze and actions in first person video (ECCV 18)
> - [2] HowTo100M: Learning a Text-Video Embedding by Watching Hundred Million Narrated Video Clips (ICCV 19)
> - [3] From Lifestyle VLOGs to Everyday interaction (CVPR 18)

---

> > ### Comment · Reviewer_zAh2 · 2021-08-25
> > **Response to rebutttal**
> >
> > Thank you for the thorough clarification and the additional experimental results, it is much appreciated! The justification for and performance of the combined agent on the interaction-light tasks alleviates some of the concerns I had, and with the proposed changes for clarity, I have adjusted my score to 7.

---

### Official Review · Reviewer_HGQh · 2021-07-16

**Rating:** 6
**Confidence:** 4

**Summary:**

This paper introduces an approach to discover activity-context priors, that is, for a given object, the environment states that are preconditions for attempting given activities with objects. Such priors are acquired from in-the-wild egocentric videos that are captured from a camera that is worn by people performing daily activities. A visual model is trained to detect how humans prepare their environment for their activities from egocentric videos. Video-based priors are then used as auxiliary rewards to encourage agents to seek out similar activity-context states. The proposed approach accelerates learning and generalises to unseen environments.


**Ethical Concerns:**

No ethical concerns.

**Limitations And Societal Impact:**

The authors adequately addressed limitations and edge cases. The work does not have societal impact is such.

**Main Review:**

Originality: The paper presents a method that is based on the insight that while humans and embodied agents have very different action spaces and bodies, they operate in similar environments. Hence, instead of tackling the gap and transfer between embodiments, the goal of the authors here is to guide agents' interactions towards states that are "demonstrated" by humans and that enables meaningful object interactions. The tasks and methods introduced have limited novelty, and the work is a novel and interesting combination of well-known techniques and ideas. It is clear how this work differs from previous contributions, and related work is adequately cited.

Quality: The submission is technically sound and claims are supported by theoretical and experimental analysis. The proposed method and the alternative methods used as comparison are used in an appropriate way. Comparisons with other existing methods are analysed across different experiments and tests, which completes the evaluation of the proposed method. Limitations of the proposed method are also discussed, and failure cases are presented.

Clarity: The paper is clear and well organised. Details about the implementation and algorithms used are provided, so that enough information is available to reproduce the results.

Significance: The results presented are significant and interesting, the ideas at the basis of this work are valuable and can be used by other researchers in the field to improve performance on complex tasks that entail sequences of object interactions. The idea of guiding agents' interactions towards states that enable meaningful object interactions from human demonstration is not completely novel, but its realisation in this work is neat.

Other comments:
Would the learned visual model work on real robot/real environments as well? Can you elaborate on the simulation vs. reality gap?
Videos are a great addition to the submission as they make it easier to appreciate the behavior of the proposed method w.r.t. others.


**Time Spent Reviewing:**

3

---

> ### Author Response · Authors · 2021-08-10
> **Response to Reviewer HGQh**
>
> Thank you for the helpful feedback.
>
> ---
> > The idea of guiding agents' interactions towards states that enable meaningful object interactions from human demonstration is not completely novel, but its realisation in this work is neat ... The tasks and methods introduced have limited novelty, and the work is a novel and interesting combination of well-known techniques and ideas.
>
> We appreciate the comments that the "realisation in this work is neat", and that we present a "novel and interesting combination of well-known techniques".  We are not sure what parts of the idea are being referred to as "not completely novel" and what aspects of the tasks/methods have "limited novelty." The review does not point to any work that already studies the problem beyond the ones we already cite and contrast to in L104-14. To the best of our knowledge, our idea to incorporate human activity priors for exploration of object interactions is novel, and has not been attempted before. More generally, as mentioned in L89-96, prior work has used human demonstration primarily for imitation learning, while our work is the first to translate free-form, real-world human video into embodied agent skills.
>
> ---
> > Would the learned visual model work on real robot/real environments as well? Can you elaborate on the simulation vs. reality gap?
>
> Our paper is motivated by the eventual goal of letting real-world robotic agents directly benefit from human video.  There are key sim-to-real challenges involved beyond standard gaps in visual quality.  (1) Our approach leverages any object class overlap between the simulation and video. Mapping AI2-THOR objects to EPIC objects is non-trivial and prone to errors (see Sec 3.3 in the main paper and Sec S4 in Supp). (2) Tasks in simulation  naturally have certain restrictions compared to real world activities. Humans perform complex object manipulations which are difficult to simulate. Despite these challenges, the sim-to-real problem is an active topic of research that is seeing exciting progress. We believe that in reasonably similar environments (e.g. kitchens in our case), with strong object detectors, and robots equipped with decent manipulation capabilities, our ideas have the potential to work in real world environments.

---

> > ### Comment · Reviewer_HGQh · 2021-09-10
> > **Thank you for your reply**
> >
> > Thank you for addressing my comments.

---

### Official Review · Reviewer_dhvp · 2021-07-16

**Rating:** 8
**Confidence:** 4

**Summary:**

In this paper, the authors train a vision model to learn an activity context from unlabeled video, use this activity context to derive a reward augmentation strategy, and train a model-free algorithm (PPO) to bring compatible objects (matching the distribution of objects scene during training) together.

**Ethical Concerns:**

Possibly with respect to biases in the video data, not sure.

**Limitations And Societal Impact:**

There are a few areas that if addressed may increase the score of my review.

The paper could benefit from internal ablations, as it is unclear the contributions of each component of the model to overall performance. I would be interested to see how replacing individual components of the model (ResNet18 backbone, LSTM aggregator) contribute to the overall model performance.

One limitation is that although the activity-context model derives from unlabeled video data, the model itself relies heavily on priors learned from a dataset that includes a large (100K) subset of labeled data [2]. It would be beneficial to see if an action recognition model trained explicitly on the Epic Kitchens dataset only showed strong generalization.

On the topic of generalization, the authors make the claim that the method generalizes to unseen environments. I think it’s important to note the distinction that these activity context priors all share a common underlying distribution (drawn from common household kitchen objects). One relevant ablation would be to alter the distribution of objects to preserve object categories, but such that the distribution no longer matched the distribution of a kitchen.

The authors do not directly address societal impact in the paper. As noted, the Epic Kitchens Dataset is anonymous and publicly available. It may be the case that as data-driven approaches to learning contexts on unsupervised data evolve, ethical concerns will arise about the nature of entrained biases.


**Main Review:**

* Originality
The paper's novel contribution is threefold. The authors propose a novel method for learning an embedding of activity priors for video, the authors use these priors to create a reward augmentation strategy, and the authors show that when trained with this strategy, pre-existing model-free algorithms are able to outperform SOTA on interaction heavy tasks.  Learning an embedding of activity priors is not, in itself new [1]. As noted by the author's memory-based methods for reinforcement learning, are also, not in themselves, novel [2]. The authors' primary contribution is reformulating the representation of the memory in terms of activity context, and deriving the reward augmentation from this representation.

* Quality
The paper shows strong results against four competing algorithms (NavExp, IntExp, ScenePriors). The results use both real-world image data and a realistic simulation environment. The figures are clear, and support the author’s central claim about the utility of activity context priors in driving the object aggregation behavior.

* Clarity
The paper is clear, the authors describe the formulation of the activity context and the adaption of the activity context to the reward. The description of how to generate and store the activity context, and how this is then used by the agent policy is also clear.

Figure 4. was a bit hard to parse, the contrast of the auxiliary reward against the background made it difficult to interpret, and the blue lines (which may represent the link of the supporting activity context to a given auxiliary reward) were not explained in the caption.

* Significance
Video datasets are abundant in comparison to supervised datasets of agents performing home activities. Developing methods that can learn useful priors from unlabeled videos would be an extreme boon to the embodied learning community. This work presents an early step towards that goal, as it does a dataset with a comparatively narrow distribution of objects and features (kitchen) Furthermore, this work presents a novel reward augmentation strategy derived from the learned object-centric priors.


**Time Spent Reviewing:**

8

---

> ### Author Response · Authors · 2021-08-05
> **Preliminary clarification questions about the review**
>
> Thank you for the helpful feedback! We will provide a complete response to the concerns above soon, but we had some preliminary questions about parts of the review that we hoped the reviewer could clarify.
>
> ---
> > Learning an embedding of activity priors is not, in itself new [1]. As noted by the author's memory-based methods for reinforcement learning, are also, not in themselves, novel [2].
>
> What are [1] and [2] here? The review does not mention which papers these refer to.
>
> ---
> > It would be beneficial to see if an action recognition model trained explicitly on the Epic Kitchens dataset only showed strong generalization.
>
> We are not sure in what way an action recognition model can be used directly for the current task. Does the reviewer suggest using an observation encoder (i.e. ResNet) that is pre-trained on EPIC action classification? Or something else?
>
> ---
> Clarifications on these points would help us put together more complete responses to the review. Thanks!

---

> > ### Comment · Reviewer_dhvp · 2021-08-10
> > **Re: Preliminary clarification questions about the review**
> >
> > Sorry, my bad, forgot to include the references!
> > [1] Social Activity Recognition on Continuous RGB-D Video Sequences, Coppola et al. '19
> > [2] Generalizable Episodic Memory for Deep Reinforcement Learning, Hu et al.

---

> > > ### Author Response · Authors · 2021-08-12
> > > **Re: References [1] and [2]**
> > >
> > > Thanks for the pointers to the papers!
> > >
> > > About [1]: The paper addresses human social activity detection (e.g. hand shake, hug) in streaming video. The "activity priors" (called proximity priors in Sec 6.3) encode distance features between humans, to produce a prior probability distribution over activity classes. This is different from our "activity priors" which entails scoring object-object relationships in the context of human-object interaction, and then using these priors for policy learning (rather than for activity detection). We do believe that our idea to incorporate human activity priors for exploration of object interactions is novel.
> > >
> > > About [2]: The paper presents a parametrized memory model to better group episodes together in episodic memory based RL methods. The memory structure studied here is different from the memory we consider (object locations relative to other objects + compatibility scores), and the goal is different as well (they use it for sample efficiency in episodic memory RL, we use it to support our reward function). Similar memory structures to ours have been used in RL (e.g., semantic top-down maps [A], observation memory [B]) however, our use case is novel.
> > >
> > > - [A] Object Goal Navigation using Goal-Oriented Semantic Exploration (NeurIPS 20)
> > > - [B] Scene memory transformer for embodied agents in long-horizon tasks (CVPR 19)

---

> > > > ### Comment · Reviewer_dhvp · 2021-08-25
> > > > **Response to rebuttal**
> > > >
> > > > Thank you for the clarifications and the additional experiments. I have adjusted my score

---

> > ### Comment · Reviewer_dhvp · 2021-08-22
> > **Thank you for the clarifications!**
> >
> > Dear authors,
> > I really appreciate the effort in replying to my questions and the ablation study.
> >
> > With respect to my question that could not be understood: the authors use a pretrained active object detection model [3] and faster-rcnn model (from EPIC kitchens). The question was to examine the performance gain (or loss) from switching from a pretrained model to a model solely trained in the kitchen environment used in the simulation. This suggestion was to determine if generalization performance was derived from the pretrained model, and the dependence of the method on having a large labeled object dataset. This suggestion can be dropped.
> >
> > [3 ]Dandan Shan, Jiaqi Geng, Michelle Shu, and David F Fouhey. Understanding human hands in contact at internet scale. In CVPR, 2020.

---

> ### Author Response · Authors · 2021-08-10
> **Response to Reviewer dhvp**
>
> Thank you again for the detailed feedback. Pending the requested clarifications in our earlier post on 08/05/21, we address the remaining points below.
>
> ---
> > Learning an embedding of activity priors is not in itself new [1]. As noted by the author's memory-based methods for reinforcement learning, are also, not in themselves, novel [2]
>
> As mentioned in our previous request for clarification, the review does not mention what [1] and [2] refer to (we are quite sure that the reviewer did not mean the papers we cite as [1] and [2]). To the best of our knowledge, our idea to incorporate human activity priors for exploration of object interactions is novel, and has not been attempted before. More generally, as mentioned in L89-96, prior work has used human demonstration primarily for imitation learning, while our work is the first to translate free-form, real-world human video into embodied agent skills.
>
> ---
> > It would be beneficial to see if an action recognition model trained explicitly on the Epic Kitchens dataset only showed strong generalization.
>
> As mentioned in our previous request for clarification, we do not follow this suggestion.  It is not clear how a video action recognition model can be used directly for the current task of learning policies for multi-step behaviors. Moreover, training such a model requires action class labels, which is not required in our approach (L148).
> If the reviewer could clarify the question, we'd be happy to elaborate.
>
> ---
> > Figure 4. was a bit hard to parse, the contrast of the auxiliary reward against the background made it difficult to interpret, and the blue lines (which may represent the link of the supporting activity context to a given auxiliary reward) were not explained in the caption.
>
> The figure compares two methods to highlight: (1) the training signals available to agents and (2) how agent behavior changes over time as a result. IntExp agents receive uniform rewards for new interactions, regardless of how well aligned they are with human goals. As a result, there is only one avenue to increase rewards — perform more interactions. In contrast, our agents are rewarded non-uniformly depending on relevant ACOs (blue lines), and can thus increase their reward in two ways — perform more interactions as before, but also transport objects intelligently to enhance the value of the interactions (larger dot size). Optimizing for this accelerates training of our agents (Fig. 5, left). We will make this clearer in L293-9, and mention the blue lines in the Fig 4 caption.
>
> ---
>
> > I would be interested to see how replacing individual components of the model (ResNet18 backbone, LSTM aggregator) contribute to the overall model performance.
>
> Thanks for the suggestion. We ran experiments comparing different backbones (ResNet18 vs. ResNet50) and aggregation modules (LSTM vs. GRU) for both our model and the baselines. We evaluate on the unseen test episodes for 4 interaction-heavy tasks. The average results of 2 training runs are in the table below.
>
> |           |          |    R18 + | LSTM     |          |   |          |    R50 + | LSTM     |          |   |          |    R18 + | GRU      |          |
> |-----------|----------|---------:|----------|----------|---|----------|---------:|----------|----------|---|----------|---------:|----------|----------|
> |           | *Cool*   | *Store*  | *Heat*   | *Clean*  |   | *Cool*   | *Store*  | *Heat*   | *Clean*  |   | *Cool*   | *Store*  | *Heat*   | *Clean*  |
> | *Vanilla* | 0.07     | 0.00     | 0.02     | 0.29     |   | 0.04     | 0.00     | 0.01     | 0.26     |   | 0.31     | 0.03     | 0.03     | 0.38     |
> | *NavExp*  | 0.02     | 0.02     | 0.01     | 0.44     |   | 0.02     | 0.00     | 0.00     | **0.42** |   | 0.00     | 0.00     | 0.03     | 0.35     |
> | *Ours*    | **0.30** | **0.16** | **0.11** | **0.55** |   | **0.15** | **0.11** | **0.12** | 0.36     |   | **0.52** | **0.31** | **0.17** | **0.73** |
>
> Using stronger backbones seems to help marginally, but does not offer conclusive results. Using the simpler GRU based aggregation (instead of LSTM) results in large improvements. Overall, the trends remain consistent across all configurations: Vanilla < NavExp < Ours. Architectural changes alone in the baselines (to either the backbone, or the aggregation mechanism) are not enough to compensate for task difficulty — performance on Cool, Store and Heat remain low (<10%) for Vanilla and NavExp.
>
> ---
> > On the topic of generalization, the authors make the claim that the method generalizes to unseen environments. I think it’s important to note the distinction that these activity context priors all share a common underlying distribution (drawn from common household kitchen objects). One relevant ablation would be to alter the distribution of objects to preserve object categories, but such that the distribution no longer matched the distribution of a kitchen.
>
> Our experiment results in Tables 1 and 2 highlight generalization to new, unseen kitchen environments (unique floor plans, object positions, and different object textures). This demonstrates generalization within the domain scope. It is correct that object classes are drawn from similar distributions. Altering that distribution is an interesting thought. However, the downstream tasks require objects from the "kitchen distribution". Each of our episodes randomizes the number and position of objects from this distribution, but if it is changed dramatically (e.g. replace kitchen objects with bathroom objects), key objects may be missing and the task will be ill-defined, which would affect all methods negatively. For example, pots/pans do not spawn in bathrooms, but they are required for the Heat and Prep task.
>
> ---
> > The authors do not directly address societal impact in the paper. As noted, the Epic Kitchens Dataset is anonymous and publicly available. It may be the case that as data-driven approaches to learning contexts on unsupervised data evolve, ethical concerns will arise about the nature of entrained biases.
>
> Our goal was to answer the question: "how can embodied agents learn from human experience?". The models may inherit the biases and statistics of the video datasets they are trained on. The activity context priors learned may capture relationships that are not always present in the real world (e.g., not all households have dishwashers or ovens), which should be kept in mind when deploying systems in the real world.

---

### Official Review · Reviewer_htnr · 2021-07-28

**Rating:** 8
**Confidence:** 4

**Summary:**

The paper presented a novel method that extracts object priors from egocentric videos to guide the learning of a robotic agent to interact with objects. The key idea is to model the presence of objects and their co-occurrence from naturalistic activities in first person videos (so called activity context), which can be further used as an auxiliary reward in RL for learning to perform complex object manipulation tasks. The proposed method was demonstrated in a simulated robotic learning environment, with promising results across several tasks.

**Limitations And Societal Impact:**

Yes.

**Main Review:**

**Originality**

Teaching a robotic agent by learning from first person visual experience is a quite exciting idea. It is great to see the first successful demonstration of such an idea.

The design of activity context using statistics of object presence and co-occurrence is smart, avoiding the potentially complicated modeling of human body motion. The idea of using object prior as auxiliary reward in RL is interesting, though the RL part is quite standard.

Overall, this is a well executed paper with a great idea.

**Quality**

The proposed method is technically sound. The experiments are solid and the results are promising. I appreciate the video demo included in the supplement.

My only concern is that all results are reported under simulation. It will be quite interesting to see some demonstrations in a real world setting. This is less of a concern but more of a hope. I do believe that the ability to demonstrate this idea in the real world could be quite impactful.

**Clarity**

The paper is well written and easy to follow. The key idea is clearly illustrated with nice visuals across the paper.

The authors claimed that “the model did not use the interaction / action label” in L148. Yet L242 suggests that the experiment considered 40K clips cropped from the video, where the starting and ending time of these clips are given by the action annotations. I think this part should be clarified.

**Significance**

Teaching robotic agents using real world videos is a critical problem in computer vision and robotics. The success of this paradigm can potentially have a major impact in robotic learning.

**Post-rebuttal Update**

Thanks for the authors' response. After reading other reviews and responses, I remain quite positive about this paper. Thus, I stand by my previous rating and recommend to accept this paper.


**Time Spent Reviewing:**

3

---

> ### Author Response · Authors · 2021-08-10
> **Response to Reviewer htnr**
>
> Thank you for the encouraging and valuable feedback.
>
> ---
>
> > It will be quite interesting to see some demonstrations in a real world setting. This is less of a concern but more of a hope. I do believe that the ability to demonstrate this idea in the real world could be quite impactful.
>
> Yes - we completely agree! Our paper is indeed motivated by the eventual goal of letting real-world robotic agents directly benefit from human video. This is reflected in our choice of tasks — realistic home robot assistant settings (L254-5) — and choice of simulator domain — kitchens, a common indoor environment that humans regularly use (L244-6). The key challenge here is of course the sim-to-real gap in terms of visuals, simulation fidelity, and object distributions. This is an active area of research that is seeing exciting progress, and we do believe that our work is a first step towards reaching the real-world transfer goal.
>
> ---
> > The authors claimed that “the model did not use the interaction / action label” in L148. Yet L242 suggests that the experiment considered 40K clips cropped from the video, where the starting and ending time of these clips are given by the action annotations. I think this part should be clarified.
>
> That's right. We use clip boundaries to denote when an action occurs, but do not use information about what action it is (action class label). We will clarify this in L148. Though we do not explore this, class agnostic clip boundaries could be detected automatically using various heuristics such as hand/object motion thresholds or tracking hand-object contact.

---

### Decision · Program_Chairs · 2021-09-27

**Decision:**

Accept (Spotlight)

**Comment:**

This paper considers an interesting problem of aiming to use egocentric human videos as a prior for guiding the behavior of an embodied agent completing tasks in AI2-iTHOR virtual environment. All of the reviews are positive about the paper, and the author response also helped address several points of feedback from the reviewers. The authors are strongly encouraged to incorporate the feedback and the changes mentioned in the author response into the revised paper.